

# Observations of gas-phase products from the nitrate radical-initiated oxidation of four monoterpenes

Michelia Dam[1], Danielle C. Draper[1], Andrey Marsavin[2], Juliane L. Fry[2], and James N. Smith[1]

[1]Department of Chemistry, University of California, Irvine, USA
[2]Chemistry Department & Environmental Studies Program, Reed College, Portland, USA

**Correspondence:** James N. Smith (jimsmith@uci.edu)

**Abstract.** Chemical ionization mass spectrometry with nitrate reagent ion ($NO_3^-$ CIMS) was used to investigate the products of nitrate radical ($NO_3$) initiated oxidation of four monoterpenes in laboratory chamber experiments. $\alpha$-Pinene, $\beta$-pinene, $\Delta$-3-carene, and $\alpha$-thujene were studied. The major gas-phase species produced in each system were distinctly different, showing the effect of monoterpene structure on the oxidation mechanism and further elucidated the contributions of these species to particle

formation and growth. By comparing groupings of products based on ratios of elements in the general formula $C_wH_xN_yO_z$, the relative importance of specific mechanistic pathways (fragmentation, termination, radical rearrangement) can be assessed for each system. Additionally, the measured time series of the highly oxidized reaction products provide insights into the ratio of relative production and loss rates of the high molecular weight products of the $\Delta$-3-carene system. Measured effective O:C ratio of reaction products were anti-correlated to new particle formation intensity and number concentration for each system;

however, monomer:dimer ratio of products was positively correlated. Gas phase yields of oxidation products measured by $NO_3^-$ CIMS correlated with particle number concentrations for each monoterpene system, with the exception of $\alpha$-thujene, which produced a considerable amount of low volatility products but no particles. Species-resolved wall loss was measured with $NO_3^-$ CIMS and found to be highly variable among oxidized reaction products in our stainless steel chamber.

## 1 Introduction

The largest uncertainty in modern climate models is attributed to the radiative effect of aerosols (Stocker et al., 2013). Their chemical complexity makes it challenging to predict their formation as well as properties that determine their direct and indirect impacts on climate. A significant fraction of total global aerosol is secondary organic aerosol (SOA), which are formed from the oxidation of gas-phase volatile organic compounds (VOCs) to form highly oxidized species that may partition into particles or small clusters (Kroll and Seinfeld, 2008; Ehn et al., 2014; Bianchi et al., 2019). Many SOA formation pathways have been

widely studied, such as the ozone ($O_3$) and hydroxyl radical (OH) initiated oxidation of biogenic volatile organic compounds (BVOCs) (Berndt et al., 2016; Lee et al., 2006; Atkinson and Arey, 2003). One such system that has been shown to contribute significantly to SOA formation, but has not been as comprehensively studied, is nitrate radical-initiated ($NO_3$) oxidation of BVOCs (Ng et al., 2017). $NO_3$ radical, produced by the oxidation of nitrogen dioxide ($NO_2$) with $O_3$, is mainly anthropogenic in origin and is most abundant at night when photolysis does not occur (Brown and Stutz, 2012). BVOCs are emitted naturally



by plants and comprise a large fraction of global VOCs, but BVOC concentrations are highest in forested regions (Acosta Navarro et al., 2014). Therefore, $NO_3$-initiated oxidation of BVOCs is an SOA-generating system that couples anthropogenic oxidants with biogenic precursors. This chemistry has also been shown to be important in areas like the southeastern United States (Ayres et al., 2015) and the Colorado Rocky Mountains (Fry et al., 2013).

Monoterpenes (MT), unsaturated $C_{10}H_{16}$ compounds, comprise a large fraction of global BVOCs and have been shown to have a high SOA production potential from nitrate radical-initiated oxidation (Ng et al., 2017; Sindelarova et al., 2014; Ayres et al., 2015). However, the large range in SOA yield in laboratory studies of the most abundant monoterpenes, $\alpha$-pinene (0-16%), $\beta$-pinene (27-104%), $\Delta$-3-carene (68-77%) (Ng et al., 2017; Fry et al., 2014; Boyd et al., 2015; Hallquist et al., 1999), indicates that the oxidation mechanisms of these MTs have key differences. As $\alpha$-pinene is often used as the representative MT in regional and global models, these oxidation mechanisms need further investigation to improve model predictions of SOA yield from MT + $NO_3$ systems and concomitant impacts on climate.

Recently, computational and experimental studies have shed light on the initial steps of $NO_3$-initiated oxidation of $\alpha$-pinene, $\beta$-pinene and $\Delta$-3-carene, all of which are bicyclic monoterpenes with a single double bond (Kurtén et al., 2017; Draper et al., 2019). These studies concluded that, following $NO_3$ addition onto the carbon-carbon double bond and rapid $O_2$ addition to the alkyl radical, first generation peroxy radical reactions are too slow to contribute to overall oxidation product distributions and instead rapidly reduce to alkoxy groups through bimolecular reactions with $NO_3$, $HO_2$ or $RO_2$ in the nighttime atmosphere. In the oxygen-rich atmosphere ($O_2$ concentration > $10^{15}$ cm$^{-3}$), $O_2$ addition to nitroxy-alkene radical compounds is expected to be much faster than radical decomposition (Berndt and Böge, 1995). First-generation alkoxy scissions play an important role in determining potential for further radical propagation for these monoterpenes and may help explain why the $\alpha$-pinene + $NO_3$

system produces much lower SOA yields than $\beta$-pinene + $NO_3$.

First generation alkoxy scissions are affected by the position of the endocyclic or exocyclic double bond with respect to the secondary ring in these bicyclic monoterpenes (Vereecken and Peeters, 2009). The most favorable, lowest energy, first generation alkoxy scission pathway for the $\alpha$-pinene + $NO_3$ system leads to the formation of pinonaldehyde, a closed shell species that is not very highly oxidized and thus not expected to contribute to the formation of new SOA. This contrasts the

$\Delta$-3-carene and $\beta$-pinene systems, which form alkyl radicals that allow for further radical propagation, oxidation, and internal isomerization. These processes can lead to the formation of highly oxidized gas-phase products that can readily partition into small particles. Additionally, other unimolecular processes have been shown to be competitive on the time scale of these reactions, including internal hydrogen-shift isomerization and radical rearrangement by opening the secondary ring (Vereecken and Peeters, 2010). The size of the secondary ring strongly influences the energy barrier for ring opening: four-membered rings

($\alpha$-pinene, $\beta$-pinene) are unlikely to open but strained three-membered rings ($\Delta$-3-carene) are much more susceptible to ring opening (Kurtén et al., 2017). Understanding the prevalence of these early unimolecular processes is important in determining the potential for further radical propagation and oxidation.

For this experimental study, we investigate $NO_3$-initiated oxidation of four monoterpenes, $\alpha$-pinene, $\beta$-pinene, $\Delta$-3-carene, and $\alpha$-thujene, in a reaction chamber. The first three monoterpenes are abundant in the atmosphere (Sindelarova et al., 2014)

and their oxidation mechanisms have been previously studied in laboratory experiments and theoretical computational studies.





$\alpha$-Thujene, a key component of frankincense oil, is less naturally abundant, but studying the $\alpha$-thujene system presents a unique opportunity to assess early unimolecular processes because of its structure. $\alpha$-thujene has a three-membered secondary ring, similar to $\Delta$-3-carene, with an adjacent double bond position, similar to $\alpha$-pinene. Following NO$_3$ addition to the double bond, oxidation of the alkyl radical to a peroxy radical and subsequent reduction to an alkoxy radical through a bimolecular

reaction is expected to occur. The first generation alkoxy scission pathways available for $\alpha$-thujene mirror those of $\alpha$-pinene (Figure 1). Cleaving the top methyl group or the left carbon bond leading to an unstable alkyl radical on a three-membered ring would be unfavorable. Cleaving the right carbon bond generates an alkyl radical on the nitrate-substituted carbon, which would lead to rapid radical termination with loss of NO$_2$, forming thujenaldehyde. If, instead, NO$_3$ addition onto the double bond was followed by radical rearrangement by opening the three-membered ring and subsequent tertiary alkyl radical formation, radical

propagation pathways become available and can potentially lead to further oxidation and condensable species (Vereecken and Peeters, 2009). By studying the oxidation of $\alpha$-thujene in addition to the previously studied MT systems, we can assess the prevalence of this ring opening reaction in the early mechanism.



**Figure 1.** Scheme of proposed $\alpha$-thujene + NO$_3$ oxidation mechanism. Orange arrows indicate formation of thujenaldehyde, a volatile product that is not expected to contribute to new particle formation. Downward arrow indicates a potential alkyl radical rearrangement that leads to a product that can undergo additional oxidation to form highly oxygenated molecules (HOMs).

This study is an observation of detailed compositional differences of observed gas phase oxidation products as they relate to the current understanding of oxidation mechanisms using high resolution time-of-flight chemical ionization mass spectromertry

(HR-TOF CIMS) with NO$_3^-$ reagent ion. NO$_3^-$ CIMS has been used to measure composition of oxidized organics in laboratory studies (Berndt et al., 2015; Rissanen et al., 2014; Hyttinen et al., 2015; Mentel et al., 2015; Riva et al., 2019) and in ambient air (Ehn et al., 2014) but has also been used specifically to probe nitrogen-containing oxidized monoterpenes (Draper et al., 2019). The NO$_3^-$ reagent ion has been shown to cluster with highly oxidized compounds that contain at least two hydrogen bond donor sites. Therefore, we don't expect to be able to measure highly volatile aldehyde products (thujenaldehyde/pinonaldehyde) with

NO$_3^-$ CIMS, but we can probe formation of highly oxidized products and compare the differences in composition among the four MT systems. NO$_3$ radical-initiated oxidation of $\alpha$-thujene has not been previously studied and so the results of this particular system are unique observations. Iodide (I$^-$) has been used extensively in previous studies as a CIMS reagent ion,





specifically to measure oxidized organics and inorganic nitrogen species (Brophy and Farmer, 2015; Aljawhary et al., 2013; Lee et al., 2014; Lopez-Hilfiker et al., 2016). The use of both $I^-$ and $NO_3^-$ reagent ions could provide additional mechanistic information and is planned for future studies.

## 2 Experimental methods

We ran chamber experiments using a darkened 560 L stainless steel chamber in flow-through mode with a total flow of 17 lpm, resulting in a residence time of ∼33 minutes. A schematic of our experimental set-up is shown in Figure 2. All experiments were performed under dry conditions. We generated $NO_3$ radical by oxidizing $NO_2$ with $O_3$ inside the chamber and allowing the oxidants to reach steady state (∼2 hours) before adding MT. $O_3$ was generated by UV photolysis of air scrubbed of $NO_x$ (NO+$NO_2$) and VOCs by a zero air generator (Aadco Instruments, model 737-13) at 1.5 lpm. $NO_2$ was introduced directly

from a commercially prepared cylinder (Praxair, Inc., 2.5 ppm in purified air) at 1.5 lpm. Before addition of MT, the mixing ratios of nitrogen compounds in the chamber were as follows: $[O_3] \approx 240$ ppb, $[NO_2] \approx 240$ ppb, $[NO_3] \approx 0.2$ ppb, $[N_2O_5] \approx 25$ ppb. $O_3$ was measured using an absorption gas analyzer (2B Technologies, model 106-L). $NO_2$ was measured with both an absorption gas analyzer (2B Technologies, model 405nm) and with a home-built thermal-dissociation–cavity ring-down spectrometer (TD-CRDS, (Keehan et al., 2020)). $NO_3$ radical and $N_2O_5$ concentrations were modeled using the kinetic box

model, KinSim (Peng and Jimenez, 2019), which was run on the Igor Pro computing platform (Wavemetrics, Inc., version 7). $N_2O_5$ was also measured by $I^-$ CIMS and the TD-CRDS, but those measurements were performed primarily to estimate wall losses and were thus not calibrated. We introduced MT into the chamber from gas cylinders that were prepared by injecting liquid MT (supplementary information Sect. SI 0.1) into the cylinders and then pressurizing with ultra-pure nitrogen. The mixing ratio inside the cylinders (11-20 ppb) was estimated using the mass of injected liquid and confirmed using a gas

chromatograph with flame ionization detector and a home-built cryogenic preconcentrator (GC-FID). The small MT flow (25 ccm) was fed into the center of a larger zero air flow (1 lpm) with tee fitting to more effectively carry it into the chamber. Monoterpene concentrations (∼41 ppb) inside the chamber were estimated from the flow dilution and concentrations, were confirmed using GC-FID, and were modeled with KinSim. The remainder of the flow, 12 lpm, was zero air that was introduced into the chamber with a Teflon "shower head" consisting of a capped tube with holes drilled perpendicularly along the length

of the tube to facilitate mixing in the chamber. We ran experiments under continuous flow and measured precursor and product concentrations for 1-2 hours until the gas-phase products reached steady state, as detected with $NO_3^-$ CIMS.

The TD-CRDS was used to measure nitrogen-containing species (nitric acid, alkyl nitrates (ANs), peroxy nitrates (PNs), $NO_2$) in both gas and particle phases (Keehan et al., 2020). The TD-CRDS measured total ANs + PNs during this experiment, and the $NO_2$ channel was used to parameterize the KinSim model. Additionally, a scanning mobility particle sizer (SMPS),

which consisted of a differential mobility analyzer (TSI, Inc., model 3081), condensation particle counter (CPC) (TSI, Inc., model 3020), and a home-built flow and voltage controller, was used to measure the particle number-size distributions. We probed the formation of low volatility oxidation products using $NO_3^-$ CIMS, consisting of a high resolution time-of-flight mass spectrometer (Tofwerk AG, model LTOF) operating in V-mode. We used a home-built transverse ionization CIMS inlet,





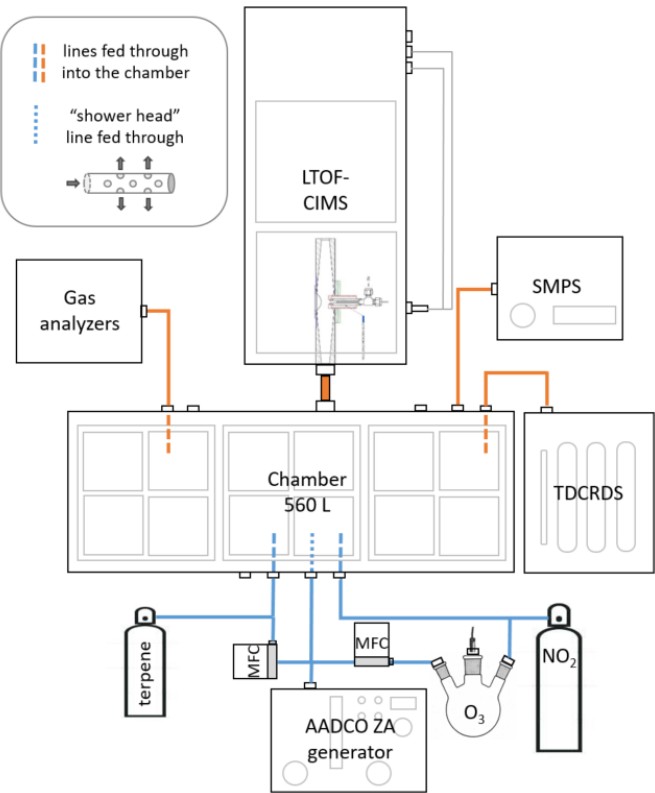

**Figure 2.** Experimental set up for chamber studies. Blue lines indicate flow going into the chamber and orange lines indicate where flow is being removed.

in which the sample flow is perpendicular to the flow of reagent ions into the entrance orifice of the mass spectrometer (Li et al., 2019). The inlet minimizes wall losses of sampled gases and clustering with neutral compounds such as water vapor in the ion source, with an average ionization reaction time of ∼80 ms. $NO_3^-$ reagent ion was generated by flowing ultrahigh purity $N_2$ gas over the head-space of a small glass vial filled with nitric acid ($HNO_3$). The reagent ion flow (8 ccm) was diluted with a larger flow (1 lpm) of pure $N_2$. Peaks in mass spectra were fitted and assigned using the Tofware software (Aerodyne Research, version 7). The reagent ion monomer ($NO_3^-$), dimer ($HNO_3NO_3^-$) and trimer ($HNO_3(NO_3)_2^-$) were used as calibration peaks for the low mass range. High mass calibrations were determined from the highest intensity single peaks in the monomer and dimer region clustered with $NO_3$ ion, and reasonable formulas were predicted from the base MT formula ($C_{10}H_{16}$).

## 2.1 Kinetic modeling

10    KinSim was used to support our experiments by providing approximate concentrations of unmeasured oxidant species ($NO_3$/$N_2O_5$) and helping assess the dominant oxidant chemistry ($NO_3$ vs $O_3$) in the chamber. The rate constants for all reactions considered in the model are listed in SI Sect. 0.1. The chamber is assumed to be well-mixed in the model and dilution flow is also





represented. Experimentally measured time series of oxidants agree well with the modeled concentrations (Figure 3). Experimentally measured wall loss of $N_2O_5$ ($1.5 \times 10^{-3}$ $s^{-1}$) was also considered in the model. This value was experimentally determined by stopping the flow of oxidants to the chamber and making up the lost flow with additional zero air, therefore "turning off" the chemistry, and using $I^-$ CIMS to measure the decay of the $N_2O_5I-$ cluster (SI Sect. 0.5). The raw $N_2O_5I^-$

decay was exponentially fitted to determine the wall loss rate. Even though excess $O_3$ remains in the chamber from the generation of the $NO_3$, the model predicts that more than 98 % of oxidation products in all MT systems should be initiated by $NO_3$. This is expected, since the rate constant for the MT + $NO_3$ radical reaction is several orders of magnitude faster than for MT + $O_3$. The rate constants of $NO_3$ and $O_3$ oxidation for $\alpha$-thujene are unknown and therefore cannot be modeled in the same way, so we estimated a "worst case scenario" by taking the slowest $NO_3$ + MT rate constant ($\beta$-pinene, $2.5 \times 10^{-12}$ from Ng et al.

(2017)) and the fastest $O_3$+MT rate constant ($\alpha$-pinene, $8.0 \times 10^{-12}$ from Khamaganov and Hites (2001)). This resulted in a ratio of $O_3$:$NO_3$ products of 0.1.

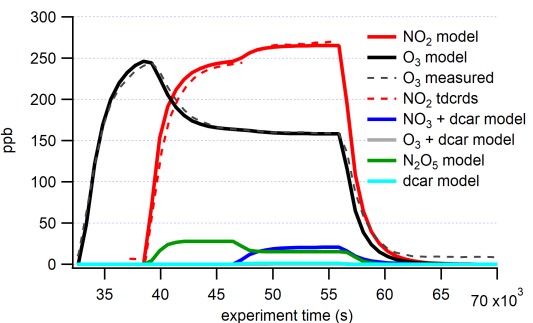

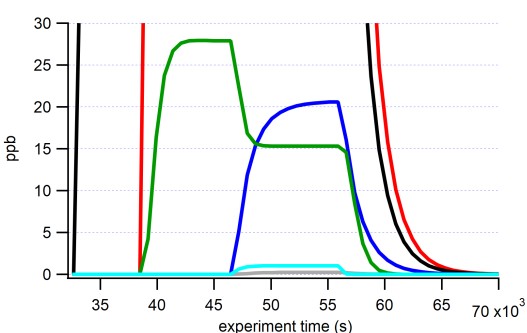

**Figure 3.** KinSim simulation of the $\Delta$-3-carene + $NO_3$ experiment. Measured traces are shown with dashed lines and modeled results are shown with solid lines. Additional modeling of the other MT systems are shown in the supplemental information (Figure S1).

## 3  Results and discussion

### 3.1  Oxidation product quantification

The total gas-phase product concentration was estimated using the sum of the abundance of all species detected by $NO_3^-$ CIMS,

integrated over the duration of the experiment. Integrated ion signal was converted to concentration using a calibration factor ($6 \times 10^{10}$ molecules cm$^{-3}$). This calibration factor was determined experimentally using, as a proxy, the reaction of sulfuric acid ($H_2SO_4$) and $NO_3^-$ ion, which is estimated to be at the collision limit as detailed by Kurtén et al. (2016). A calculation is shown in the SI in Sect. 0.3. We recognize that this calibration factor results is likely an upper limit for the actual concentration of organic nitrate compounds. Mixing ratios of reaction intermediates calculated from integrated concentrations were on the

order of 0.1 to 0.001 ppt. Percent yield is reported for each MT system: $\Delta$-3-carene ($2.5 \times 10^{-4}$) > $\alpha$-thujene ($1.0 \times 10^{-4}$) > $\beta$-pinene ($5.0 \times 10^{-5}$) > $\alpha$-pinene ($9.8 \times 10^{-6}$).





Substantial wall losses were found for $NO_3^-$ CIMS-measured species, importantly, with high variability observed among individual species and no clear trend with O:C or molecule weight (see SI Sect. 0.5). Therefore, individual wall loss corrections were applied for all species before calculating yield from $NO_3^-$ CIMS measurements. Explicit wall effects were only measured for the $\Delta$-3-carene system, so for the other MTs, average wall loss rates for monomers ($5.5 \times 10^{-3} \ s^{-1}$) and dimers ($3.4 \times 10^{-3}$

$s^{-1}$) were calculated using observations from the $\Delta$-3-carene system data and are thus subject to greater uncertainty. In contrast, wall losses of the total ANs+PNs measured by TDCRDS were observed to be negligible relative to the dilution timescale of the chamber, suggesting that the majority of this bulk organic nitrate signal is due to higher-volatility species not measured by the $NO_3^-$ CIMS, not substantially lost to the walls, and likely not contributing substantially to SOA formation. The trend in observed molar yields of total ANs+PNs is consistent with previous measurements ($\Delta$-3-carene > $\beta$-pinene > $\alpha$-pinene), but

the magnitudes (20%, 10%, 5%) are substantially lower than previous studies. This is puzzling, but may be due to the TDCRDS measurement measuring only the subset of high-volaility nitrates in these experiment, while the lower-volatiliy nitrates have rapid wall losses that prevent the TDCRDS from measuring them. We note that the CIMS is essentially inlet-less in comparison to a 2 m inlet line for the TCRDS in these experiments.

The $\alpha$-thujene system produced more highly oxidized products than the $\alpha$-pinene system, and especially significant is the

amount of highly oxidized products that were formed from $\alpha$-thujene. The $\Delta$-3-carene system generated the most particles, followed by the $\beta$-pinene system, and both the $\alpha$-pinene and $\alpha$-thujene systems did not generate any particles (Figure S2). Observed particle number trends agree with gas-phase product trends except for the $\alpha$-thujene system, in which highly oxidized products formed but did not nucleate and/or grow effectively to form measurable particles.

### 3.2   Comparison of oxidation product composition

#### 3.2.1   Definition of categories for elemental analysis

The experiment-averaged mass spectra for each MT system showed very different peak distributions (Figure 4). Here, we explore the relative ratios of experiment-averaged ion abundance from categories of products. For comparison, we normalized the integrated area of each peak by the total integrated area of all organic peaks in the mass spectrum. The general formula $C_wH_xN_yO_z$ was used to create categories of reaction products that correspond to specific predicted mechanistic pathways that

are summarized in Table 1, with representative mechanisms for each pathway shown in Figure 5. Carbon number provides an indication of fragmentation caused by C-C bond cleavage for any carbon number that is not equal to 10 (for monomers) or 20 (for dimer formation from peroxy or alkoxy radical additions, e.g. $RO_2 + RO_2$ or RO+RO). Hydrogen number informs us about the terminal functional group and associated bimolecular reactions leading to them. Nitrogen number can indicate secondary double bond generation or $NO_2 + RO_2$ chemistry from residual $NO_2$ in the chamber. Structure activity relationships

and rate constants were found in literature for relevant pathways (Vereecken and Peeters, 2009, 2010; Novelli et al., 2021; Kurtén et al., 2017; Draper et al., 2019; Jenkin et al., 2019; Crounse et al., 2013), but proposed mechanisms were not explicitly modeled for this study. Oxygen number and O:C ratio provide insights into the number of generations of oxidation chemistry that occur. While we cannot completely explain the origin of every observed compound, the observed differences provide

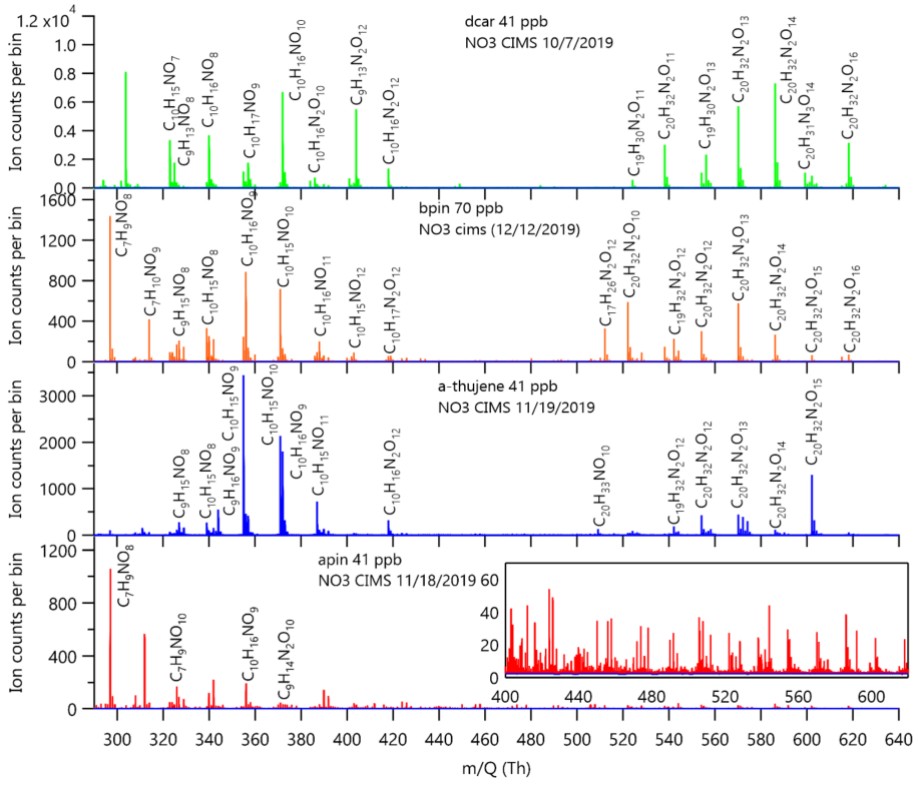

**Figure 4.** Stacked mass spectra from each MT + $NO_3$ system. $NO_3^-$ reagent ion is included in assigned formula. Left axes are raw ion counts and not normalized to the reagent ion.

valuable information regarding the most important pathways that lead to these low volatility products. The ratio distributions for all the MT systems are summarized in Figure 6. Complete peak lists for each MT system can be found in the SI (Sect. 0.7). Note that the $\beta$-pinene system ratios are an average of an 8 ppb experiment and a 70 ppb experiment (39 ppb average). The full comparison of results from both experiments can be found in the SI (Sect. 0.6).

### 3.2.2 Categorized monomer composition and mechanistic implications

The relative abundance of dimers compared to monomers (Figure 6) correlates with observed particle formation (SI Sect. 0.2) for each MT system, with the exception of $\alpha$-thujene, for which we observed substantial dimer formation without significant particle formation. We predict that the first alkyl radical formed from $NO_3$ addition onto the $\alpha$-thujene double bond can potentially shift to form a new double bond while opening the strained three-membered ring, forming a six-membered ring and new tertiary alkyl radical (Figure 1). That structure can then undergo further oxidation to form highly oxygenated, lower volatility compounds, including dimers. Our observations of highly oxygenated products for $\alpha$-thujene shows that, under these





**Table 1.** Summary table of reaction product formulas grouped by mechanistic pathways shown in Figure 5. Group (a) details formation of $C_{10}H_XN_1O_Z$ reaction products, group (b) details formation of $C_9H_XN_1O_Z$ products, group (c) details formation of $C_7H_XN_1O_Z$ products and group (d) details various pathways for dimer formation.

| Group | Formula | Functional groups | Pathway |
|---|---|---|---|
| A | $C_{10}H_{16}NO_z$ | $RO_2$ | $MT + NO_3$ |
| | $C_{10}H_{17}NO_z$ | ROH,ROOH | $RO_2+H$, $RO+H$ |
| | $C_{10}H_{15}NO_z$ | R=O | RO-H |
| B | $C_9H_{13}NO_z$ | CHO + R=O | $MT-CH_3$ |
| | $C_9H_{14}NO_z$ | CHO + $RO_2$ | $MT-CH_3$ |
| | $C_9H_{15}NO_z$ | CHO + OOH | $MT-CH_3$ |
| C | $C_7H_9NO_z$ | R=O + R=O | $MT-C_3H_6$ |
| | $C_7H_{10}NO_z$ | R=O + $RO_2$ | $MT-C_3H_6$ |
| | $C_7H_{11}NO_z$ | R=O + ROH,ROOH | $MT-C_3H_6$ |
| D | $C_{20}H_{32}N_2O_z$ | ROOR | $C_{10}H_{16}NO_z + C_{10}H_{16}NO_z$ |
| | $C_{20}H_{33}N_3O_z$ | ROOR + ROH,ROOH | $C_{10}H_{16}NO_z + C_{10}H_{17}N_2O_z$ |
| | $C_{19}H_{30}N_2O_z$ | ROOR | $C_{10}H_{16}NO_z + C_9H_{14}NO_z$ |
| | $C_{17}H_{26}N_2O_z$ | ROOR | $C_{10}H_{16}NO_z + C_7H_{10}NO_z$ |

experimental conditions, there is an additional pathway for the first generation alkoxy scission (Figure 1) that is competitive with the formation of thujenaldehyde.

    **Carbon number**. Figure 6 summarizes the monomer carbon number distributions, which provide insights into fragmentation pathways. Detailed schemes for each MT system are shown in the SI Sect. 0.4. In general, alkoxy decomposition pathways

($C_7$, $C_9$) are predicted to be competitive with H migration when leaving groups become highly substituted (Novelli et al., 2021; Vereecken and Peeters, 2010). For three of the MT systems studied ($\Delta$-3-carene, $\beta$-pinene, $\alpha$-thujene), terminal sites that are are available become oxidized and lead to fast alkoxy decomposition. For the $\alpha$-pinene system, however, such a terminal site is not available. Another possible pathway for fragmentation is alkyl radical rearrangement that leads to ring opening. For $C_7$ compounds, radical rearrangement creates new alkyl radicals in tertiary isopropyl sites. Oxidation of the alkyl radical site to an

$RO_2$ radical and decomposition to an RO radical allows alkoxy decomposition to occur, generating a new alkyl radical (Kurtén et al., 2017). For $C_9$ compounds, ring opening and $NO_3$ radical addition creates a new alkyl radical alpha to a $CH_3ONO_2$ group. If the alkyl radical is again oxidized (forming $RO_2$) and decomposed (forming RO), alkoxy scission with an $ONO_2$ group on the alpha carbon would not have a smaller energy barrier than a site with OH or OOH alpha substitution, but this barrier would still be smaller than an alkyl-substituted site.

We were surprised to see a negligible $C_7$ contribution for the $\Delta$-3-carene system because substantial $C_7$ contribution was observed in Draper et al. (2019) for a similar experiment with the same instrument. That study used a commercial $NO_3^-$ CIMS inlet (Aerodyne Research, model TOF CIMS) that features a longer reaction time between analyte and reagent ion ($\approx$0.15







**Figure 5.** Scheme of mechanistic pathways listed in Table 1. Potential bimolecular reaction partners in a nighttime atmosphere ($NO_3$, $RO_2$, and $HO_2$) are abbreviated as X. Panel (a) shows $\Delta$-3-carene oxidation as an example of how hydrogen number can indicate radical termination. Panel (b) shows a $C_9$ fragment formation pathways available to the $\Delta$-3-carene and $\alpha$-thujene systems. Panel (c) shows two potential $C_7$ fragment formation pathways available to the $\beta$-pinene and $\alpha$-pinene systems.

s), which might contribute to a different detected ion distribution compared with the transverse ionization inlet used here. We plan future studies to explore the source of these differences. A large $C_7$ contribution was observed for both the $\alpha$-pinene and $\beta$-pinene systems. One possible formation mechanism is shown by the R1 arrow in Figure 5C, in which the initial alkyl radical formed from $NO_3$ addition rearranges to open the four-membered ring, forming a new tertiary alkyl radical. This early alkyl

radical rearrangement was proposed by Boyd et al. (2015) for the $\beta$-pinene + $NO_3$ system. However, $O_2$ addition to the initial alkyl radical is expected to be fast in our experimental conditions and in the ambient atmosphere (Berndt and Böge, 1995; Kurtén et al., 2017), so we expect this alkyl radical rearrangement not to be significant. R2 in Figure 5 shows an alkyl radical rearrangement that opens the four-membered ring and generates a new tertiary alkyl radical. This specific ring opening reaction is not currently supported by modeling, so we cannot currently comment on its competitiveness. Another possibility is that this

$C_7$ compound comes from the left-side first-generation alkoxy scission pathway (R3 in Figure 5C). While predicted to be a minor pathway for $\alpha$-pinene, this left side scission is expected to be equally, if not more, competitive with the other two first generation alkoxy scissions (top and right) for $\beta$-pinene (Claflin and Ziemann, 2018). This branching ratio may be somewhat reflected in the relative yields observed from the $NO_3^-$ CIMS. The $C_7$ product yield from the $\alpha$-pinene system is roughly half that of the $\beta$-pinene system. This early left-side scission makes a secondary radical on the four-membered ring, which is less

stable than an acyclic analog because of ring strain (Kurtén et al., 2017). In abundant $O_2$, this alkyl radical will be oxidized to an $RO_2$ radical. Because this new $RO_2$ radical is confined by the rigid structure of the ring, autoxidation could be slow, while





| | δ-car | β-pin | α-pin | α-thuj |
|---|---|---|---|---|
| **monomers** | 50% | 60% | 81% | 66% |
| **dimers** | 50% | 40% | 19% | 34% |
| $C_7$ | 1% | 27% | 55% | 1% |
| $C_8$ | 0% | 0% | 0% | 1% |
| $C_9$ | 27% | 7% | 10% | 15% |
| $C_{10}$ | 72% | 66% | 35% | 83% |
| $C_{17}$ | 1% | 22% | 6% | 1% |
| $C_{18}$ | 2% | 0% | 14% | 2% |
| $C_{19}$ | 13% | 18% | 12% | 32% |
| $C_{20}$ | 84% | 60% | 68% | 65% |
| $N_0$ | 3% | 1% | 17% | 3% |
| $N_1$ | 79% | 83% | 57% | 91% |
| $N_2$ | 18% | 16% | 25% | 5% |
| $H_{15}$ | 28% | 48% | 45% | 77% |
| $H_{16}$ | 57% | 45% | 48% | 19% |
| $H_{17}$ | 15% | 7% | 7% | 4% |
| **% yield** | $10^{-4}$ | $10^{-5}$ | $10^{-6}$ | $10^{-7}$ |

Row group labels (left margin): monomer carbon ($C_7$–$C_{10}$); dimer carbon ($C_{17}$–$C_{20}$); $C_{10}N_x$ ($N_0$–$N_2$); $C_{10}NH_Y$ ($H_{15}$–$H_{17}$)

**Figure 6.** Ratios of reaction products separated into categories for each MT system. Each category for each MT system adds up to 100%. The color axis indicates magnitude of total % yield, which can be compared across the MT systems. $\beta$-pinene percentages are averaged from a high (70 ppb) and low (8 ppb) mixing ratio experiments, but the rest of the MT systems are at the same mixing ratio (41 ppb).

in the presence of other radical species ($NO_3$, $RO_2$), bimolecular decomposition from a peroxy radical ($RO_2$) to an alkoxy radical (RO) could be faster. Few cyclic $RO_2$ H migration computational studies exist for relevant systems (Kurtén et al., 2015; Xu et al., 2019; Draper et al., 2019), but bimolecular rate constants with $NO_3$, $NO_2$ and $HO_2$ have recently become better defined for $RO_2$ radicals (Jenkin et al., 2019). Once the RO radical is formed, it is possible that decomposition of a C-C bond,

5    thus opening the strained ring and leading to formation of a ketone and a tertiary alkyl radical, could be fast, with a predicted energy barrier of 0.6 kcal mol$^{-1}$ from structure-activity relationships (SARs) (Vereecken and Peeters, 2009, 2010; Novelli et al., 2021). In this position, the tertiary radical resembles the structure predicted for $\Delta$-3-carene in Kurtén et al. (2017) and could lead to the loss of the isopropyl group and formation of a $C_7$ fragment. This pathway is only available for the $\beta$-pinene and $\alpha$-pinene systems and significant contribution from $C_7$ compounds is only observed from these two systems.



Instead of abundant $C_7$ compounds, we observe a large contribution from $C_9$ compounds for the $\Delta$-3-carene system and smaller $C_9$ signal for the rest of the MT systems. A possible pathway for $C_9$ formation is through the generation of a secondary double bond and subsequent $NO_3$ addition (Figure 5B). The $CH_3ONO_2$ leaving group could quickly fragment into $CH_3O$ and $NO_2$ (Novelli et al., 2021; Vereecken and Peeters, 2010), but is assumed to be more stable than a $CH_3$ radical formed from

any other terminal site on the molecule in this case. The most abundant $C_9$ compound for the $\Delta$-3-carene system contains two nitrogen atoms ($C_9H_{14}N_2O_{12}$). From our predicted pathway, the resulting product may contain one or two nitrogen atoms because even if the second 2 group is expected to be part of the leaving group, a radical site remains on the molecule, making the $RO_2 + NO_2$ pathway possible via reaction with residual 2 in the chamber. For the $\alpha$-thujene system, a $C_9$ compound is predicted to be formed in an analogous pathway to the $\Delta$-3-carene system, as terminal double bonds may be generated. For

the $\alpha$-pinene system, $C_9$ formation would be possible if an alkyl radical rearrangement occurred on the four-membered ring, generating a terminal double bond. Finally, for the $\beta$-pinene system, a terminal site is available after the initial right-side alkoxy scission, but alkyl radical ring opening could also be possible if a left-side scission were to occur.

**Hydrogen number**. As previously mentioned, hydrogen number is an indication of the bimolecular termination pathway and the presence of the subsequent terminal functional group for $RO_2$ and $RO$ radicals. For $C_{10}$ compounds with a single nitrogen

atom (Figure 5A), $H_{15}$ closed shell compounds indicate aldehyde groups created from hydrogen abstraction on a carbon atom alpha to an alkoxy radical. $C_{10}H_{16}NO_z$ compounds are radicals and most likely peroxy-radical compounds, as the lifetimes of alkoxy and alkyl radicals are very short. $C_{10}H_{17}NO_z$ compounds are closed shell and contain hydroxy or hydroperoxy terminal groups formed from abstracting a hydrogen atom from a different molecule. $C_{10}H_{16}NO_z$ radical compounds were the most abundant for the $\Delta$-3-carene, $\beta$-pinene and $\alpha$-pinene systems. In contrast, $C_{10}H_{15}NO_z$ aldehyde/ketone compounds were the

most abundant for the $\alpha$-thujene system and and mainly distributed among two compounds ($C_{10}H_{15}NO_9$ and $C_{10}H_{15}NO_{10}$). If one assumes that the concentration of bimolecular reaction partners is similar for each MT system, it appears that the reaction products generated in the $\alpha$-thujene system included structures with especially labile hydrogen atoms next to a possible alkoxy radical site. In that case, this bimolecular hydrogen abstraction reaction would be fast and lead to a high ratio of $C_{10}H_{15}NO_z$ compounds relative to the $C_{10}H_{16}NO_z$ and $C_{10}H_{17}NO_z$ compounds. Additionally, a moderate $C_{10}H_{15}NO_z$ contribution was

observed for the $\beta$-pinene system from the same formula ($C_{10}H_{12}NO_{10}$).

Analogously, $C_9$ radical compounds with a single nitrogen atom will contain 14 hydrogen atoms if they are formed from the loss of a methyl group, while closed shell $C_9$ compounds will contain 13 or 15 hydrogen atoms (Figure 5B). For the $\beta$-pinene system, hydroperoxy or hydroxy $C_9H_{15}NO_z$ compounds were most abundant. The dominant $C_9$ species in the $\alpha$-thujene system was a $C_9H_{16}NO_z$ species; it is not currently clear how $C_9H_{16}NO_z$ products can be formed. Surprisingly, the most abundant $C_9$

compounds for the rest of the MT systems do not fall into this $C_9H_xNO_z$ category, although some of these types of compounds make up a fraction of the overall $C_9$ signal. $C_9$ dinitrogen compounds were the most abundant for the $\Delta$-3-carene ($C_9H_{14}N_2O_{12}$) and $\alpha$-pinene ($C_9H_{14}N_2O_{10}$) systems, with 14 hydrogen atoms indicative of the formation of peroxy radical intermediates or an $RO_2NO_2$ group. Finally, $C_7$ radical compounds with a single nitrogen atom are predicted to have 10 hydrogen atoms if formed by the loss of an isopropyl group, while closed shell $C_7N_1$ compounds will have 9 or 11 hydrogen atoms (Figure 1C). The most

abundant $C_7$ compound ($C_7H_9NO_8$), observed for both the $\beta$-pinene and $\alpha$-pinene system, was closed shell and its formula is





consistent with the mechanism we present in Figure 5C, but a large radical contribution from $C_7H_{10}NO_9$ was also measured in the $\beta$-pinene system.

**Nitrogen number**. Monomer compounds with a single nitrogen dominate all MT systems, but with varying abundance. Initial addition of $NO_3$ radical leads to the formation of these single nitrogen-containing compounds. The $\alpha$-pinene system
had much higher contributions from $N_0$ and $N_1$ compounds compared to the other MT systems, but it is expected that the majority of products detected for the $\alpha$-pinene system come from minor pathways, so unusual pathways that are normally considered to be slow can potentially be competitive and lead to the observed ratios of nitrogen-containing compounds. We are not currently certain what those pathways are. The $\alpha$-thujene system had a 10% greater contribution from single nitrogen-containing compounds compared to the $\Delta$-3-carene and $\beta$-pinene systems, and a corresponding 10% smaller contribution
from dinitrogen compounds. $\alpha$-Thujene and $\Delta$-3-carene both contain strained three-membered rings that can make a ring opening alkyl radical rearrangement reaction faster than the MTs with less strained four-membered rings ($\beta$-pinene,$\alpha$-pinene) (Kurtén et al., 2017). Therefore, it is possible that a measured compound containing a single nitrogen could have made and lost a secondary double bond in the oxidation process in addition to an $NO_2$ group, instead of that single nitrogen atom being attributed to the initial $NO_3$ radical addition to the parent $\alpha$-thujene molecule. This pathway would be available early in the
oxidation mechanism for $\alpha$-thujene, but $\Delta$-3-carene requires one generation of oxidation to pass before the secondary double bond can be generated (Sect. 1). Additionally, the $\Delta$-3-carene system has other pathways that can lead to highly oxidized species whereas the $\alpha$-thujene system is currently predicted to only have one pathway that leads to secondary double bond generation. Dinitrogen compounds can be formed via $NO_2 + RO_2$ (from excess 2 in the chamber) and this can also form a product with two nitrogen atoms. The rate constant for this $NO_2 + RO_2$ reaction is highly uncertain, making explicit kinetic
modeling of the $RO_2$ fate challenging.

### 3.2.3   Categorized dimer composition and mechanistic implications

**Carbon number**. $C_{20}$ compounds were the most abundant dimers across all MT systems. However, the $\beta$-pinene system had a large (22%) $C_{17}$ contribution, whereas the other MT systems produced negligible (<6%) $C_{17}$ dimers. Correspondingly, a large $C_7$ monomer contribution was observed for the $\beta$-pinene system and not for the $\Delta$-3-carene or $\alpha$-thujene systems. $C_{19}$ dimers
were produced in all MT systems, but the $\alpha$-thujene system had $\sim$ 20% greater fractional $C_{19}$ contribution than $\beta$-pinene (10%) and $\Delta$-3-carene (18%), even though the $\Delta$-3-carene system produced the highest percentage of $C_9$ monomers. Dimers with carbon numbers other than 20 imply that after alkoxy decomposition and fragmentation of a $C_{10}$ monomer, the newly generated monomer fragment must contain an active radical site. This can occur through the generation of a secondary double bond within one of the monomer units through alkyl radical rearrangement. Another possibility is that the smaller leaving
group is a closed shell species, leaving the larger fragment with an alkyl radical. A possible example of this rearrangement is shown in Figure1C, in which a $C_7$ radical is generated with acetone as the leaving group in the $\beta$-pinene system. That $C_7$ alkyl radical can be oxidized and possibly form an adduct with a $C_{10}$ radical, making a $C_{19}$ dimer that is uniquely observed for the $\beta$-pinene system. $C_{19}$ dimers can potentially be formed through a similar radical rearrangement pathway discussed above in Sect. 3.2.2. It is possible to create a $C_{19}$ dimer from that $C_9$ fragment, as another active radical site could exist on the molecule





after NO$_3$ radical addition to the newly generated double bond. An analogous C$_9$ and C$_{19}$ dimer could be formed from the $\alpha$-thujene and $\beta$-pinene systems.

**Hydrogen number**. Closed shell C$_{20}$ compounds formed from two C$_{10}$H$_{16}$ compounds (RO$_2$ + RO$_2$) have 32 hydrogen atoms and were the most abundant type of dimers in all MT systems. If C$_{20}$ compounds have an alternate hydrogen number, it

is assumed that a second double bond was generated at some point during the oxidation process, forming a monomer species with both an active radical site and a terminated site (CHO, OOH, or OH). C$_{19}$N$_2$H$_{30}$ compounds make up ~80% of total observed C$_{19}$ signal from the $\Delta$-3-carene system and are predicted to form via the addition of a C$_{10}$H$_{16}$NO$_z$ radical and a C$_9$H$_{14}$NO$_z$ radical. In contrast, the $\beta$-pinene system produced C$_{19}$H$_{32}$N$_2$O$_z$ compounds with the highest abundance among C$_{19}$ species, and the $\alpha$-thujene system produced both C$_{19}$H$_{30}$N$_2$O$_z$ and C$_{19}$H$_{32}$N$_2$O$_z$ compounds with equal intensity. As with

the C$_9$ monomer units, it is uncertain where the extra two hydrogen atoms are gained. For the $\beta$-pinene system, C$_{19}$H$_{26}$N$_2$O$_z$ molecules were the most abundant C$_{19}$ species, which is consistent with a C$_7$H$_{10}$NO$_z$-RO$_2$ monomer fragment combining with a C$_{10}$H$_{16}$NO$_z$-RO$_2$ monomer.

**Nitrogen number**. For C$_{20}$ dimers, a nitrogen number of two can correspond to addition of RO or RO$_2$ radicals with one substituent ONO$_2$ group each. It is expected that products without nitrogen atoms lose them in termination steps. Thus, dimers

with two nitrogens cannot be formed by the addition of a dinitrogen monomer and a monomer with no nitrogen atoms unless two radical sites are available on the same molecule and one participates in RO$_2$+RO$_2$ adduct formation while the other leads to the loss of NO$_2$. This would create a monomer with no nitrogen atoms that is also bonded to another monomer unit. C$_{20}$ dimers containing two nitrogen atoms are the most abundant across all MT systems. Very small contributions are observed from N$_3$ or N$_4$ C$_{20}$ species. The $\alpha$-thujene system has a considerable contribution (13%) from single nitrogen-containing dimer species.

The single nitrogen-containing compounds can possibly be formed by the same combination of monomer units mentioned above for C$_{20}$N$_2$ compounds, except one of the monomers contains only a single nitrogen atom instead of two. Additionally, it is possible that ozonolysis-RO$_2$ and nitrate-RO$_2$ intermediates engaging in cross reactions play a role, as the rate constant for $\alpha$-thujene with ozone or NO$_3$ radical has not been measured.

### 3.3   Effective O:C ratio of oxidation products

The average, weighted O:C ratios for total organic reaction products are as follows: $\Delta$-3-carene (0.71), $\beta$-pinene (0.62), $\alpha$-thujene (0.45), and $\alpha$-pinene (0.73). The reported effective O:C ratio for all molecules only includes oxygen atoms presumed to be on the carbon backbone (two oxygen atoms subtracted for every NO$_3$ group). Even though the average O:C ratio correlates with observed particle formation for every system except $\alpha$-pinene, the effective O:C ratio distribution is very different for all systems (Figure 7). Additionally, if products are grouped into monomer and dimer species, the effective O:C ratios do

not necessarily correlate with observed particle formation. For monomer species, the average effective O:C ratios were found to be anti-correlated with the intensity of new particle formation events, with the $\Delta$-3-carene reaction products having the lowest effective O:C (0.700) followed by $\beta$-pinene (0.724), $\alpha$-thujene (0.771), and $\alpha$-pinene (0.780). The dimer products all hover around a similar range of effective O:C ratio: $\Delta$-3-carene (0.37), $\beta$-pinene (0.43), $\alpha$-thujene (0.37), and $\alpha$-pinene (0.5). In general, we observe lower O:C ratios for dimers compared to monomers. If the average O:C for the dimer compounds is





around 0.4, they were possibly formed from monomers with an average O:C of 0.6. The average monomer O:C observed from all experiments was 0.7. For the $\alpha$-thujene system, the majority of the monomer species and the dimer species are centered in a narrow effective O:C range. This is in contrast to the observations from the other MT systems. If the three-membered ring is opened early on in the oxidation mechanism, and the subsequently generated double bond is attacked by a $NO_3$ radical, the

5    molecule will be symmetrical in one plane (Figure 1). It is possible that this symmetry leads to a lack of diversity of products, reducing the possibility of structurally unique products by half. It is important to note that $NO_3^-$ CIMS is selective towards highly oxidized species, so the oxygen distribution reported here is within the limits of the sensitivity of the reagent ion. Every O:C bin is assumed to be ionized with the same efficiency.

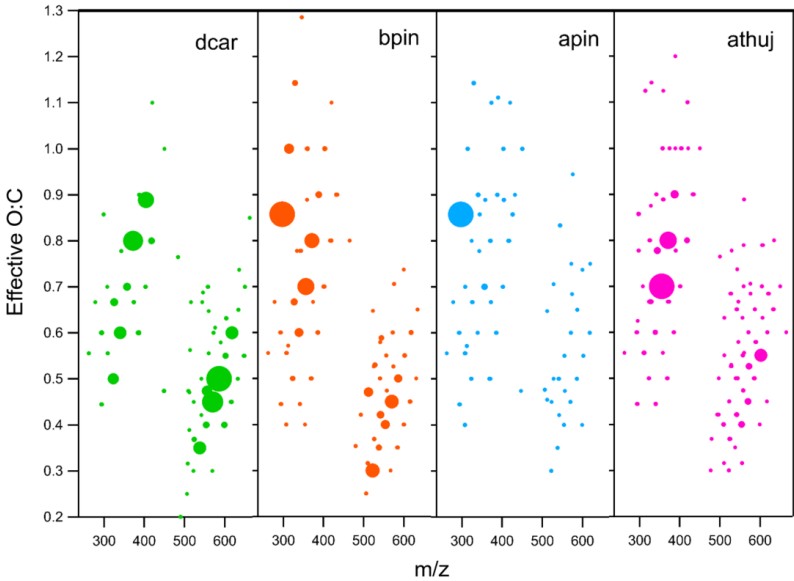

**Figure 7.** Effective O:C ratio plotted versus m/z for all measured oxidation products. Plots include oxygen atoms only on the carbon backbone. Each marker represents one compound, and marker area is proportional to signal intensity. Note that each MT system is scaled to its own maximum intensity, thus marker sizes cannot be compared across MT systems. The $\beta$-pinene plot shown is for the high mixing ratio (70 ppb) experiment, and all others are 41 ppb

### 3.3.1 Temporal analysis of oxidation products

10    When comparing the uncorrected (not corrected for wall losses) time series traces for each MT system, a decrease in product signal after the initial increase was observed for the two systems ($\Delta$-3-carene, $\beta$-pinene) that exhibited new particle formation (Figure 8). These decreases roughly correspond to increasing particle number. More quantitative analysis of particle growth rates is beyond the scope of the current study, but is planned for future studies. The decrease in gas-phase products over the course of the experiment was not observed for the MT systems that did not produce particles ($\alpha$-pinene and $\alpha$-thujene).





Time series curves of $\Delta$-3-carene + $NO_3$ products show the difference between the rates of formation and sink for these compounds. At this time, this analysis is only available for the $\Delta$-3-carene system; however, we expect similar behavior for the products of the other MT systems. By grouping the individual species into the categories detailed in Sect. 3.1, insights can be gained into the net formation time ($x_{1/2}$), which takes into account wall loss but not loss due to condensation sink on

particles. The time series curves were fitted to sigmoidal curves to determine the time it took for the signal to reach one-half the maximum ($x_{1/2}$). In general, monomers were found to have faster net formation time $x_{1/2}$ than dimers (Figure 8). $C_{10}H_{16}NO_z$ compounds have the fastest $x_{1/2}$ and are the most abundant monomer species. Most of these species are also single nitrogen-containing compounds. Within the $N_1$ category, $C_{10}H_{16}NO_z$ compounds and $C_{10}H_{17}NO_z$ compounds have similar $x_{1/2}$ times, whereas the $C_{10}H_{15}NO_z$ species have a slower $x_{1/2}$ overall, falling into the same $x_{1/2}$ regime as the dimer compounds. If

assuming these $C_{10}$ compounds all have a similar condensation sink rate, it appears that $H_{16}$ radicals and $H_{17}$ hydroxy and hydroperoxy compounds form faster than the $H_{15}$ carbonyl compounds. $C_9$ compounds have more variable $x_{1/2}$ times, but are, in general, slower than the fastest $C_{10}$ monomers. Additionally, the most abundant $C_9$ species ($C_9H_{14}N_2O_{12}$) has an $x_{1/2}$ value in the dimer region (note that this compound contains two nitrogen atoms). Additionally, for $C_{10}$ compounds with one nitrogen atom, there is an almost imperceptible increasing trend relating $x_{1/2}$ and effective O:C ratio (Figure 8b). A similar slight trend

can be observed for $C_{20}$ compounds. It is possible that, because the formation and sink times cannot be isolated, more highly oxidized molecules take longer to form but also condense more rapidly.

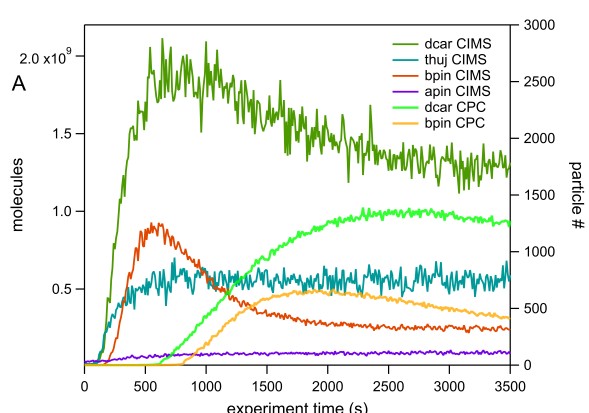
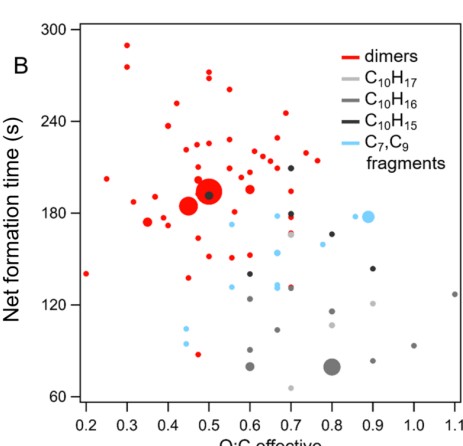

**Figure 8.** (a) Time series of total organic compounds for each MT system measured with $NO_3{}^-$ CIMS plotted with associated particle number concentration. The $\beta$-pinene traces shown are from the high concentration (70 ppb) experiment. (b) Formation:sink ratio of reaction products of the $\Delta$-3-carene system plotted against effective O:C ratio.

## 4   Conclusions and implications for atmospheric chemistry

$NO_3{}^-$ CIMS has been used to probe the composition of $NO_3$-MT oxidation products in laboratory chamber experiments in order to gain mechanistic insights. The major species formed in each system were distinctly different, showing the effect of





MT structure on the oxidation mechanism. We initially hypothesized that the structural similarities between $\alpha$-thujene and $\alpha$-pinene would lead to the dominance of relatively high-volatility oxidation products thujenealdehyde and pinonaldehyde, respectively. Our results, however, suggest that an alkyl radical rearrangement can lead to an intermediate that can undergo additional oxidation and form HOM monomers and dimers (Figure 1) in $\alpha$-thujene oxidation. The lack of measurable new
particle formation in spite of the presence of these dimers indicates a more complex relationship between HOMs and new particle formation. This should be studied in greater detail to provide insights into the ability of HOMs to participate in nanoparticle formation and growth.

For all systems, HOM carbon number, an indicator of fragmentation pathways, supports the notion that decomposition is more likely when leaving groups become highly substituted. The presence of substantial amounts of $C_7$ fragments for the $\beta$-
pinene and $\alpha$-pinene systems is consistent with the loss of an isopropyl group from those species and we have hypothesized the mechanism by which this occurs (Figure 5c). Hydrogen number for $C_{10}N_1$ compounds, an indicator of termination pathways and the presence of closed-shell or radical intermediates, show the dominance of peroxy radical $H_{16}$ compounds for all but the $\alpha$-thujene system, the latter of which was dominated by closed-shell $H_{15}$ aldehydes or ketones. For $C_9N$ products, closed shell $H_{15}$ hydroperoxy or hydroxy compounds dominated the $\beta$-pinene system and $H_{16}$ species dominate the $\alpha$-thujene system by
a mechanism that is unclear to us. For $\alpha$-pinene and $\Delta$-3-carene systems, the dominant $C_9$ compounds detected were species containing 2 nitrogen atoms and 14 hydrogen atoms. The dominant $C_7$ fragment observed for $\alpha$-pinene and $\beta$-pinene was a closed-shell $C_7H_9NO_8$ compound with possible isomers, again consistent with the mechanism we propose in Figure 5C. Nitrogen number for all monomers was dominated by those containing a single nitrogen atom, which arises from the initial $NO_3$ radical addition. Some later generation monomeric dinitrogen compounds were detected in all systems with the exception
of $\alpha$-thujene, the latter of which has fewer pathways for the formation of a second double bond. This limitation may be partly responsible for the lack of observed new particle formation for $\alpha$-thujene despite the abundance of HOMs in that system.

The observed dimers included major peaks containing 20, 19, and 17 carbon atoms, which is consistent with the observed monomers containing 10, 9, and 7 carbon atoms. Hydrogen numbers for all systems indicate that $C_{20}$ dimers form predominantly closed-shell compounds with 32 hydrogen atoms. In general, our observations of hydrogen and nitrogen number in
detected dimers are consistent with the composition of detected monomers, which suggests dimer formation by cross reactions between nitrate-containing $RO_2$ species.

Detected O:C ratios of gas-phase products provide some insights into NPF mechanisms. In general, monomer O:C ratios are inversely correlated with new particle formation intensity and dimer O:C ratios are less than those of the monomers, but generally similar for all systems ($\sim$0.4). It is likely that monomers with higher O:C ratios are preferentially partitioning into
growing nanoparticles and, indeed, we observe a decrease in HOMs coincident with an increase in the concentration of newly formed particles as discussed in Sect. 3.3.1.

Finally, our temporal analysis of oxidation products from the $\Delta$-3-carene system shows unique, species-dependent formation rates and provide insights into wall loss rates. In general, dimers formed more slowly than monomers. Since dimers had lower O:C ratios, there was a weak anti-correlation between O:C ratio and the net formation time. This correlation is not apparent
for monomers, but $C_{10}$ monomers did display some trends such as $H_{16}$ radicals and $H_{17}$ hydroxy and hydroperoxy compounds





forming faster than $H_{15}$ carbonyl compounds. Additional applications of this temporal analysis approach for the other MT systems would be an interesting extension of this work.

The information gained from this detailed comparison of gas-phase composition with currently established mechanisms provides new information on these oxidation processes and further elucidates the effect of these species on particle formation and growth. A wider range of oxidation products (SVOCs) need to be measured to observe the compounds not detected by $NO_3^-$ CIMS in order to more comprehensively draw conclusions about particle formation potential. Further analysis of particle formation rate, particle composition and modeling of energy barriers for some of the proposed mechanistic pathways is needed. Additional spectroscopy can also be useful for confirming the presence of functional groups.

*Author contributions.* MD and JS conducted chamber experiments and curated, analyzed, investigated, and visualized data from all measurements. DD, AM, and JF helped design chamber studies, conducted TDCRDS analysis and kinetic modeling. MD prepared the manuscript with contributions from DD, AM, JF and JS. JS and JF acquired funding and supervised the project.

*Competing interests.* The authors declare that they have no conflict of interests.

*Acknowledgements.* This research was supported by funding from the US National Science Foundation (NSF) under grant no. AGS-1762098 and by the US Department of Energy (DOE) under grant DE-SC0019000. The authors would like to acknowledge Emily McLaughlin Santa Maria, Mike Lawler, Sabrina Chee, Hayley Glicker, Deanna Myers, Adam Thomas and Jeremy Wakeen for their contributions to discussions regarding this project.



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
