# Peer review of "Observations of gas-phase products from the nitrate radical-initiated oxidation of four monoterpenes"

_Atmospheric Chemistry and Physics, 2021_

## Author Comment (AC1)

*We would like to thank both reviewers for their constructive comments, which we feel we have addressed as stated below. The resulting manuscript is much-improved as a result of the review process. Our responses to each point are provided below in italics. Where possible, we also provide page and line numbers for the modified text in the revised version.*

General

The manuscript investigates the oxidation of several monoterpenes by NO3. The authors selected four bicyclic monoterpenes with different ring sizes and structures, in order to reveal mechanistic information from observed product distributions. New particle formation was observed in two out of four cases, and its occurrence was related to the product distribution. The core of the analysis focused on product distribution and suggested pathways to rationalize the observations. Overall, the manuscript is quite interesting and quite well written. The in the description of the mechanistic parts, especially in the beginning the authors could try to give more aids to the reader, e.g. by referring more often to the mechanistic schemes. The manuscript could be suited for publication in ACP. There are some minor issues which could lead to better readability and some major comments the authors should address before the manuscript could be published in ACP.

*Both reviewers recommended that improvements in organization would make it easier for readers to navigate the reaction schemes that we explore. To address this, we have edited the manuscript as follows:*

*Figures 1 and 5 were modified to include numbered carbon atoms for monoterpene structures, which were then referenced throughout the analysis text.*

*P.11, l.13-34 and p.12, l.13-14: Numbered monoterpene carbon atoms were referenced throughout the text for clarification of the mechanistic steps discussed. Informal "left" and "right" side scissions were further clarified.*

*The mechanism shown in Figure 5B was modified to include an explicit step showing radical rearrangement and opening of the three-membered ring to clarify the intended step.*

Major comments:

Mixing in the chamber, wall losses

p.4, l.6 and p.5, l12: From Figure 2, the chamber seems to be rectangular not cylindric. If so, I am wondering how perfect the mixing will be. How do you ensure (fast) mixing?

*This concern is understandable, given the unusual geometry of the chamber. We address these questions, along with those mentioned next, below.*

p.4, l.23: Do you know the typical mixing time? I am also asking, because the lifetime of N2O5 of about 11 min. (p.6, l.2) in such a metal chamber of this size appears to be quite

long. Or do you establish large gradients towards the walls? Actually, the value given in the main manuscript of 1.5E-3 s-1 differs from the value in the SI (1.25E-3 s-1, p.S6, l.5)

*Regarding the question about ideal mixing, we previously performed tests to assess mixing conditions. The following text describes these tests and has been added to the SI:*

"Experiments were conducted at 20 LPM inlet flow with CO2 added to the chamber. Following this we flushed the chamber with CO2-scrubbed zero air. A Licor CO2 analyzer measured [CO2]. The plot below shows the results along with the best fit line for the decay of CO2. We then fit the data to an exponential equation for dilution in a continuously stirred chamber in order to calculate the average residence time at this flow rate. The result at this flow rate, 1657 s, is to be compared to the theoretical residence time associated with a well-mixed chamber of 1680 s. This close correspondence is found at flow rates ranging from 10 - 60 lpm and confirms that our use of Teflon "shower heads" to introduce gasses into the chamber creates satisfactory mixing in the chamber."

**Characterization of residence time**

[Figure]

Figure S9: Comparison of experimentally determined residence time to that calculated assuming a well-mixed chamber.

*Regarding the discrepancy in the reported wall loss rate, 1.25E-3 s$^{-1}$, is the correct value and this has been updated in the text.*

*Regarding the mixing time, the answer is approximately 100 s. This value arises from experiments performed in which a pulse of CO2 was introduced into the chamber at t = 0 s, after which we observed the mixing process and decay of CO2. The figure below shows that it takes approx 100 s for the CO2 to reach its peak value, which is equal to the theoretically predicted value based on the concentration of CO2 added and the dilution ratio.*

[Figure]

Figure S10: Measurements of [CO2] exiting chamber after a pulse of CO2 was added at t=0.

*In order to refer the reader to this discussion and to inform them of the main conclusions of these tests, the following modification was made to the first line in the Experimental section:*

"We ran chamber experiments using a darkened 560 L stainless steel chamber in flow-through mode with a total flow of 17 lpm. Previously, we confirmed that operating in this manner results in conditions in which the chamber was well-mixed after ~100 s with a residence time of ~33 min (see SI Section 0.5)."

*Regarding the effects of potential gradients from sampling near the chamber walls, we agree that this could be an issue close to the walls but all of our sample lines extend into the chamber in order to minimize the possibility of sampling in this gradient. The following text was added to the end of the Experimental section:*

"The sample lines extended into the chamber ~20 cm in order to minimize the possibility of sampling in a gradient caused by loss of low-volatility species to the chamber walls."

p.7, l.2: SI Section 0.5 does not really show the wall loss as a function of O:C. It discusses only the range of wall loss coefficents. The example shown in Figures S7 indicates more a wall equilibrium, because the wall loss trace becomes a constant and not zero. A bit difficult to understand for the example of the heavily functionlized C20 compound. The raw signal looks as expected, though. What will happen if you fit

c(t) = exp((τ(wall) +τ(dil) x t)

and set τ(dil) to the nominal residence time?

*We agree with the sentiment expressed by the reviewer that the description for the wall loss determination procedure provided in the SI needed improvement. The expression that we used is similar to that provided by the reviewer with some small edits (we believe these are typos in the above): c(t) = c(t=0)exp[-(k_wall + k_dil) x t]. Rather than set the dilution loss rate to the nominal residence time, we determined this value directly from the decay of*

*NO2, which in a previous study was found to have a wall loss rate of 3-4 orders of magnitude lower than that of the complex organic compounds detected in this study. Since k_dil and k_wall are small compared to the time scales of reaction, we could apply a Taylor series expansion to the exponential decay terms. We note that we also checked the validity of this assumption by calculating k_wall without the Taylor expansion and achieved the same result. The relevant section in the SI was modified as follows:*

"Wall losses for individual species were measured for the dcar system by observing the time traces of ion abundance when the ion production term is zero and therefore the concentration of species A normalized by the initial concentration is determined by first order loss rates due to dilution ($k_{dil}$) and wall deposition ($k_{dep}$) as follows:

$$[A]/[A]_{o} = \exp[-(k_{dil} + k_{dep})t].$$

These conditions were created by allowing the gas phase species to reach steady state, then shutting off the oxidant and dcar flow into the chamber. The missing flow was made up with zero air in order to maintain a constant dilution rate. We used NO2 to determine $k_{dil}$, since prior experiments using the same chamber reported a wall loss rate of this compound of ~10E-6, about 3-4 orders of magnitude higher than wall loss rates observed for oxidation products in this study. The value of $k_{dil}$ can thereby be taken from an exponential fit of the NO2 data according to:

$$[NO2]/[NO2]_{o} = \exp(-k_{dil}t).$$

The wall loss rate for each species, A, can be calculated directly from the measurements by dividing the normalized \ce{NO2} concentration by the normalized concentration of species A:

$$[[NO2]/[NO2]_{o}]\backslash[[A]/[A]_{o}] = 1\backslash[\exp(-k_{dep}t)] = \exp(k_{dep}t).$$

Since the values of $k_{dil}$ and $k_{dep}$ are small compared to the time scales of these experiments, we can make the simplifying assumption that the exponential terms can be replaced by their Taylor expansion, i.e., $e^{-x} \approx 1 - x$. With this simplification, the ratio shown above can be replaced by linear terms as follows:

$$[NO2]/[NO2]_{o} - [[A]/[A]_{o}] = \exp(k_{dep}t).$$

Figure S7 shows an example of this analysis. The figure shows the normalized decay curves for NO2 and a representative C20 compound. It also shows the difference curve, to which we fit an exponential function in order to calculate $k_{dep}$. These values of $k_{dep}$ for each detected species were used to correct the measured ion abundance for wall losses, as shown in the right plot of Figure S7. These traces were fitted to sigmoidal curves to find the net formation time of each individual compound, as shown in Figure S8."

*"*

I am also wondering, why the dilution trace (=NO2) appears to be linear. Maybe it is better to use a log scale for demonstrating the losses. (By the way, I guess the units on the y-axis of Figure S7 should be "cm-3")

*We are certain that the dilution trace from NO2 follows an exponential decay. We have confirmation of this because the calculated first-order dilution rate, 2.4E-3 s-1, is consistent with the nominal residence time determined by our prior measurements as presented above. What appears as a linear decay is simply a time period in which the rate of change of the slope is small. Regarding the y-axis of Figure S7, we note that we have changed this to normalized concentration in response to our new description of the wall loss rate determination.*

p.7, l.10: Regarding the TD-CRDS measurement. What is the molar yield of the condensable organic nitrates. I guess it is of the order of percent? The yields detected by CIMS seem to be much lower. I expect the product spectrum not to be too different compared to the previous studies mentioned. Insofar the losses in a 2 m Teflon line seem not to be too critical for non-HOM, which should be the majority. Did you calibrate such line losses?

*We cannot distinguish between condensable organic nitrates and total organic nitrates with the TDCRDS measurement at this point. Line losses were found to be negligible for the TDCRDS, as the reviewer mentioned, the majority of the measured signal comes from non-HOM species.*

The shortest lifetime of HOM is about the same as for N2O5. Can this be an indication of the typical mixing time in your chamber? Once entering the thin diffusion layer on the walls the molecules get lost? Could it be that you lose significant amounts of organic nitrates on the metal walls, with a rate close to your mixing time? How stable are functionalized organic nitrates on dry walls made of stainless steel?

*Regarding mixing, as we stated previously we have experimentally determined that mixing is relatively fast for our chamber conditions (approx. one minute). As we stated above, our sample ports extended ~20 cm into the chamber so wall gradients should not be an issue.*

*Regarding the impacts of the dry stainless steel walls on organic nitrate chemistry, we saw no evidence for this but cannot rule this out as a possibility. An experiment that would specifically address this question would need to be much simpler than those presented here, focusing on single compounds or compound classes that can be easily measured.*

C7 compounds

p.9, l.15ff: The residence time for the inlet utililized by Draper et al. (2019) was with150 ms only about a factor of two longer than yours of 80 ms. Do you think the sensitivity to C7 compounds from Δ3-carene is limited by the reaction time of cluster formation? Then it should scale with the reaction time (at same reagent ion concentration)? However, isn't it more a fast dynamic forming and breaking of the reagent ion molecule clusters?

Or do you think C7 compounds are observed by Draper et al. because they a formed in their inlet due to the longer reaction time? But then, how can you be sure that you don't have chemistry going on in your inlet, too. As said, the residence times in both inlets are not too different. (I assume, both work at ambient pressures.)

The issue of different detection of C7 compounds is actually critical. If chemistry in the ion source can shift the product distribution significantly, how can you then be sure that your product (fragment) ranking and distribution represents the situation in the chamber? Or the other way round: if the C7 compounds were not detected or lost in your inlet, then they must have been still there in the chamber, as shown by Draper et al.. However, you explain mechanistically why they must be missing. As a consequence, many your mechanistic explanations for fragmentation processes would be standing on weak foot.

Can you think of other reasons for low C7 concentrations in D-Carene in your case compared to Draper et al., 2019.

*This is a valid concern and we thank the reviewer for their comments. We were in error to suggest that the different inlets would cause the discrepancy in observed reaction products. Closer inspection of the two experimental conditions suggest that the Draper (2019) study and our study were significantly different and could result in different chemistry. The conditions of the two experiments are as follows:*

*Draper's experimental conditions: [O3] ~ 370 ppb. [NO2] ~ 200 ppb. [N2O5] ~ 50 ppb. [NO3] ~ 0.6 ppb. [dcar] ~ 50 ppb. Residence time of 23 minutes.*

*Our experimental conditions: [O3] ~ 240 ppb. [NO2] ~ 240 ppb. [N2O5] ~ 25 ppb. [NO3] ~ 0.2 ppb. [dcar] ~ 41 ppb. Residence time of 32 minutes.*

*The ratio of MT to oxidant is about a third less than Draper (2019) (we have more dcar in excess) and the residence time is ten minutes longer. Having a higher monoterpene to oxidant ratio could potentially drive the RO2 + RO2 pathway and make the C7 fragmentation pathway less competitive. Additionally, a longer residence time would allow more generations of gas-phase chemistry to occur and change the nucleation and growth of particles in the chamber, potentially causing the observed reaction products to change. In the future, we will investigate the different regimes of chemistry in greater detail. We have modified the manuscript to correct our error in analysis as follows:*

*"The conditions of the two experiments were different in that the ratio of MT to oxidant was lower in our experiment (0.004 vs 0.012) and the residence time for our experiment was longer (33 min vs 23 min). These differences can lead to different chemistry inside the chamber that could result in different observed reaction product distributions. We plan future studies to explore the source of these differences further."*

Minor

p.5, l.8 and p.6, l.14-19: I suggest to moving the calibration issues up to the Experiment section.

*We agree that this suggestion will improve the structure of the manuscript and have moved the calibration discussion up as suggested.*

p.6, l.20: These yields are extremely small or did you mean molar yields and not "percent" yields, Please, check. The same in Figure 6.

*Percent yield is correct and they are indeed small, as discussed in the text. It's a very selective reagent ion, but they are still small considering this.*

p.7, l.27: Is recombination by RO+RO really a source of dimers?

*Both reviewers pointed this out and we agree that we were in error to suggest this pathway. This pathway was removed from the text.*

p.8, l.11 - p.9, l.1: "these experimental conditions" To which conditions are you referring to?

*This sentence was included to emphasize that these results are from only one experimental condition (41 ppb MT with 0.5 ppt NO3 radical) and that ratios may change under different conditions.*

p.9, Table1: I would separate the "–" sign by spaces, now it can be misinterpreted as chemical bonds.

*It is understandable how this notation caused confusion. Table 1 was edited per the reviewer's suggestion.*

p.9, section Carbon Numbers: I suggest more often to refer to the mechanistic schemes when you explain a pathway.

Why do you use the word "alpha" instead of the Greek letter?

In parts the section contains a bit lab slang: e.g "creates a new alkyl radical alpha to" should be "in α position to". This regards the description of the molecule by top, left and right bonds, too. Wouldn't it be better to number the bonds and atoms, where needed?

*As previously mentioned, carbon numbers were added to the schematics and referenced in the text for bond cleavage reactions. The mechanism was modified to show explicit radical rearrangement steps. All instances of "alpha" were replaced with the greek letter, and all instances of "alpha" regarding position on the molecule were changed to "in α position to."*

p.10, l.8: What do mean by "not currently supported by modelling". Do you mean by theoretical kinetics?

*This language was vague and we thank the reviewer for pointing that out. We added "quantum chemical" in front of modeling.*

p.13, l.5: a-pinene: N0 is higher, but N1 is lower than in the other MT. The sum of N0 and N1 in Figure 6 is a bit lower compared to b-pinene and Δ3-carene. This not the same as described in the text.

*N1 is a typo and was changed to N2, which eliminates the discrepancy pointed out here.*

p.13, l.12f: This sentence is hard to understand.

*This sentence was indeed hard to understand and was split into two sentences and simplified to, "Therefore, it is not possible to know for certain how many nitrogen atoms have been lost by a molecule in the process of oxidation. It is not possible to attribute a product containing a single nitrogen to the initial NO3 radical addition to the parent a-thujene molecule."*

p.13, l.31. C7 + C10 should make a C17 dimer, I guess.

*Yes! This is a typo and C19 was changed to C17.*

p.14, l.26: What is about hydroperoxy groups?

*The effective O:C ratio was vaguely defined here, so the sentence was changed to, "The reported effective O:C ratio for all molecules does not include the oxygen atoms from the nitrate group."*

p.14, l.33f: Monomers show a smaller spread in O:C than the dimers, which is claimed to be similar. I am not sure if the notation "anti-correlated" to observed new particle formation is the right formulation here.

*We understand the concerns of the reviewer for the term "anti-correlated" and the conclusions about dimer trends. The sentence changed to, "found to have an opposite trend," and dimer trend observation removed in order to more accurately reflect the analysis for this subsection.*

p.16, l.1: It is not clear what you mean by "difference", between formation rate and sink. Do you want to say that different products have different time series because of different formation rates and sinks.

*Yes, to clarify the objective of this sentence, it was changed to "Different reaction products have different time series because of different formation rates and sinks, as observed in the dcar system."*

p.16, l.5: The time series of curves for Δ3-carene and b pinene in Figure 8A do not look sigmaoidal. Please explain in more detail what you did for fitting the rise times.

And related: what is the time resolution of your measurement (how many data points enter a fit? The rise times could be faster than your mixing times. What would that mean for your analysis?

Actually, isn't that type of time series analysis in contradiction to your concept to operate the chamber as a flow through reactor? Again, it depends on the mixing time, better on a small ratio of mixing time over rise time.

*The time series curves in Figure 8A are the sum of all the individual time series of each chemical species and therefore, the sigmoidal quality of the individual traces are lost. The mixing time of this chamber has been previously characterized and is shown to be approximately one minute for the flow rates used in this experiment, as discussed previously. The time resolution is 10 seconds.*

p.17, l.27ff: I think these conclusions are not really justified by the data, The variation of O:C in the monomers is not very strong. There are not sufficient observations to claim correlations. You have 4 cases, α-pinene being an exception and α-thujene not doing what is expected from the dimer fraction. One has do perform more experiments probably with either more MT or at different O:C, monomer:dimer for the same MT. You must weaken that conclusion.

*We thank the reviewer for bringing this to our attention and have modified the conclusion as follows:*

*"Detected O:C ratios of gas-phase products provide some insights into NPF mechanisms. In general, monomer O:C ratios share a very small trend with new particle formation intensity. It is possible that monomers with higher O:C ratios are preferentially partitioning into growing nanoparticles and, indeed, we observe a decrease in HOMs coincident with an increase in the concentration of newly formed particles as discussed in Sect. 3.3.1."*

Typo's and small errors

p.2, l.17: I suggest to using "nitroxy-alkyl radical"; it is more precise than "nitroxy-alkene radical"

*Yes, we agree! This sentence was changed on p.2 l.18.*

p.4, l.9: Something is missing. I guess VOC were not generated by a zero-air generator but transported into the chamber by using it. I suggest to skipping it here, because you describe it later anyhow.

*This sentence is included just to indicate what zero air is (air free from NOx and VOCs), not to elucidate how monoterpene was introduced into the chamber for the experiment.*

p.4, l.11: O3 is not a nitrogen compound?!

*Yes, nice catch! The word "oxidants" was included in this sentence.*

p.7., l14: Information is doubled in this sentence.

*There was a typo that made the information seem redundant, so the second instance of "products" was changed to "dimers" to convey the actual message of the sentence.*

p.8, caption Figure 4: I guess reagent ion was excluded from formulas assigned. Please check.

*The reagent ion is not included in assigned formulas and the caption has been updated.*

p.8, l.9: "rearranges" instead of "shift"? A bond may shift but a molecule rearranges.

*"Shift" was changed to "rearrange."*

p13, l.31: R3 in Figure 5 c?

*This whole section was changed to make the references to the mechanism more clear.*

p14, l.28: … except "for" α-pinene …

References:

l.21, p.35: DOI is double.

*Thanks, one DOI was removed.*

Supplement:

p.1, l.4: the compound(s) is(are) missing: …for ???...

*All the supplement references were broken, but they have been fixed.*

p.S8, header section 06: "b-pinene"

*This header was changed to include the greek letter β.*

General Comments

This manuscript describes an experimental study of the reactions of four monoterpenes with NO3 radicals. Experiments were conducted in a flow-through stainless steel chamber and gas-phase products were analyzed online using a chemical ionization mass spectrometer with a NO3– ion source (NO3-CIMS) and a thermal desorption cavity ringdown spectrometer for nitrates (TDCRDS). Particle size and volume concentrations were monitored with a scanning mobility particle sizer (SMPS). Kinetic modeling was employed to estimate concentrations of O3 and NO3 radicals to verify that the monoterpenes primarily reacted with NO3 radicals. Attempts were also made to estimate loss of products to the chamber walls by measuring decay rates in the absence of reaction. The results were used to identify and quantify reaction products and place them in various classes (monomers, dimers, etc.), measure elemental ratios, and develop reaction mechanisms to explain the formation of the detected products for all the monoterpenes.

I think the measurements were well done, and could provide useful insights into the products and mechanisms of these reactions. The nighttime reactions of monoterpenes with NO3 radicals are of significant current interest because of the impacts of organic nitrate formation on NOx sequestration and secondary organic aerosol formation, as well as a desire to understand how monoterpene structure influences reaction products and mechanisms.

Unfortunately, I found much of the manuscript very difficult to understand. The authors base their interpretation of the results on proposed reaction mechanisms, and that discussion encompasses most of the paper. But in their presentation, they rely too much on the text to do this without providing figures of detailed reaction mechanisms that a reader needs in order to be able to follow along. The mechanisms shown in the main body of the paper are condensed to the point that they are of little value, and those in the SI are only slightly better. The text is extremely dense and detailed, and in my opinion spends too much time attempting to explain every observation. As a result, I came away not knowing what the main points were. I strongly suggest that the authors make a major effort to narrow the discussion to the main points, and create figures that allow a reader to explicitly follow all the reaction steps discussed in the text. Since I am normally quite comfortable with VOC oxidation mechanisms, I think that unless this is done the paper will be unreadable to most people who might be interested in the topic. In light of these problems, I think the manuscript might be publishable in ACP, since the experiments are interesting and of high caliber, but not without major revisions. I provide some specific comments below, but given the overall difficulties I had understanding much of the discussion, there are large sections for which I did not provide comments.

Specific Comments

1. Page 2, line 15: Do you mean peroxy radical isomerization reactions?

*Yes! The word "isomerization" was added to the text.*

2. Page 2, line 17: Why do you quote O2 concentrations > 10E15/cm3 when they are ~10E18/cm3 in the troposphere?

*The value of 10E15 cm-3 was reported by the source listed there. It is included to emphasize that in tropospheric conditions, O2 additions will be very fast because as the reviewer stated, tropospheric O2 concentration is greater than 10E15 cm-3.*

3. Page 6, line 7–11: The rate constant for a-pinene + O3 is 8.4E-17. It is also most reasonable to use values for alkenes with similar structures, especially where the C=C bond is in the ring and a methyl group is attached, since that has a large impact on the rate constants. The a-pinene + NO3 rate constant is 6.2E12. See Atkinson and Arey, Chem Rev. 103, 4605 (2003).

*We thank the reviewer for directing us to this reference for this rate constant. The rate constant has been updated in the text and the model.*

4. Page 6, line 14–16: Is 6E10/cm3 to estimated total detected product concentration? It doesn't have the right units for a calibration factor. If so, what does this correspond to as a yield, and which experiment does it apply to? And why do all the figures except 8 report ion counts instead of concentrations?

*We have changed the figure 8 axis to ion abundance (arb) and made all other axes consistent. Figure 6 is the only figure in which explicit quantitative comparisons are important. For all other figures, comparison of intensity is equally as demonstrative as using overall yield, but less distracting because the yield numbers are not important for those aspects of the analysis. The calibration factor is calculated using the method of Kurten 2012 (mentioned in the text) and is the correct units.*

5. Page 6, line 19: What do you mean by reaction intermediates? Radicals? Which ones?

*We see how the word "intermediates" is vague and can cause confusion. "Intermediates" was changed to products for greater clarity.*

Page 7, line 27: Are you suggesting that RO + RO reactions occur? This is not possible, since RO isomerization, decomposition, or O2 reactions occur on microsecond timescales, while bimolecular RO + RO reactions would occur on second timescales or longer.

*Both reviewers have mentioned this point and we agree that this suggestion was in error. Therefore, this pathway was removed from the text.*

6. Page 7, line 32: What do you mean by number of generations? In standard usage the number of generations is the number of reactions of C=C bonds in the molecule with

NO3 radicals, so here 1 or 2 generations might be formed. Also, because the presence of NO3 in the molecule is a clear indication of NO3 addition, the N/C ratio would be a better indicator of the number of generations.

*We see that the usage of the word "generations" causes confusion and thank the reviewer for pointing this out. The sentence was changed to, "how much autoxidation chemistry occured, increasing the O:C ratio of the products." Additionally, nitrogen number will not be a good indicator of "generations" in both the definition provided by the reviewer and the definition we provide because nitrogen atoms can be lost in several reactions, as detailed in the text.*

7. Table 1: What is meant by RO + H? The only RO reactions that form ROH are H-shift isomerization.

*The RO + H reaction was suggested in error and was removed from the table.*

8. Page 9, line 10: Do mean a bimolecular reaction to form a RO radical? That would not be considered decomposition, which generally refers to a unimolecular bond cleavage and dissociation.

*Yes, thank you for bringing our attention to this. We changed "decomposition" to "bimolecular reaction to form an RO radical."*

9. Page 11, line 1: See Comment 13.
10. Page 11, line 8: Don't you mean loss of acetone?

*As the isopropyl group is referred to as such in multiple places in the document, we kept this term for the sake of consistency.*

11. Page 12, lines 14–16: I think the aldehyde in Figure 5A is more likely to be formed by an RO2 + RO2 reaction via the Russell mechanism than by an RO + O2 –> R=O + HO2 reaction, since this RO radical could isomerize much faster than the O2 reaction.

*Thank you for bringing this to our attention. The aldehyde shown in Figure 5A is just a model aldehyde, not an observed molecule. The C10H15 compounds we observe are much more highly oxidized and therefore the mechanism will not be obvious, and the alpha hydrogen abstraction is just suggested as one possibility for aldehyde formation. Per the reviewer's suggestion, we have included the possibility of RO2 + RO2 reactions forming aldehyde groups in the discussion and have cited a study performed by Hasan et al. JPC 2020 on p.12 l.24 in revised version.*

12. Page 14, line 25: I think this should be "…total detected organic…".
13. Page 16, line 19: I think this should be "…major detected species…".

*Thank you, we think these changes will make the language more precise. Both points 12 and 13 were updated in the text.*

Technical Comments

1. Page 3, line 2: thujene should be capitalized.

*This was changed in the text.*

2. Page 4, line 18: (Sect. SI??)
3. Page 5, line 12: Table ??
4. Page 6, line 4: (??)
5. Page 4, line 7: (Table??).
6. Page 6, line 18: Sect. ??
7. Page 7, line 2: (see SI Sect. ??).
8. Page 7, line 11: Should be "experiments".

*This was changed in the text.*

9. Page 7, line 16: (Figure??)
10. Page 8, line 2: SI(??)
11. Page 8, line 4: (SI Figure ??).
12. Page 8, line 6: (Figure ??).
13. Page 9, line 4: SI Sect. ??
14. Page 9, line 7: Delete "are".

*This was changed in the text.*

15. Page 13, line 16: (Sect. ??).

*The supplement references were broken in the previous version of the paper, these all have been updated.*